# A Multi-Task Joint Learning Model Based on Transformer and Customized Gate Control for Predicting Remaining Useful Life and Health Status of Tools [note 1]

**DOI:** 10.3390/s24134117

**Published:** 2024-06-25

**Authors:** Chunming Hou, Liaomo Zheng

**Affiliations:** 1Shenyang Institute of Computing Technology, Chinese Academy of Sciences, Shenyang 110168, China; houchunming@sict.ac.cn; 2University of Chinese Academy of Sciences, Beijing 100049, China; 3Shenyang CASNC Technology Co., Ltd., Shenyang 110168, China

**Keywords:** remaining useful life, wear stage, multi-task joint learning, dynamic adaptive, transformer encoder

## Abstract

Previous studies have primarily focused on predicting the remaining useful life (RUL) of tools as an independent process. However, the RUL of a tool is closely related to its wear stage. In light of this, a multi-task joint learning model based on a transformer encoder and customized gate control (TECGC) is proposed for simultaneous prediction of tool RUL and tool wear stages. Specifically, the transformer encoder is employed as the backbone of the TECGC model for extracting shared features from the original data. The customized gate control (CGC) is utilized to extract task-specific features relevant to tool RUL prediction and tool wear stage and shared features. Finally, by integrating these components, the tool RUL and the tool wear stage can be predicted simultaneously by the TECGC model. In addition, a dynamic adaptive multi-task learning loss function is proposed for the model’s training to enhance its calculation efficiency. This approach avoids unsatisfactory prediction performance of the model caused by unreasonable selection of trade-off parameters of the loss function. The effectiveness of the TECGC model is evaluated using the PHM2010 dataset. The results demonstrate its capability to accurately predict tool RUL and tool wear stages.

## 1. Introduction

As a crucial component of computer numerical control (CNC) machines, the health status of tools significantly impacts machining quality. Accurately predicting the remaining useful life (RUL) and wear stage of a tool can provide valuable information to operators, enabling them to make timely adjustments to machining parameters to ensure the tool’s optimal operation. This approach can reduce production downtime and maintenance costs [1,2].

At present, the RUL prediction methods of cutting tools can be broadly categorized into two groups: physical-based methods and data-based methods [3,4,5]. The physical-based methods aim to realize the RUL prediction of the cutting tool by constructing mathematical or physical models that characterize the relationship between the performance degradation process of the tool and its cutting parameters. However, these methods require researchers to possess specialized knowledge in relevant fields, and they often rely on specific assumptions and simplifications that may not generalize well to diverse cutting conditions, resulting in poor generalization performance. In contrast, data-based methods, including statistical methods, machine learning methods, and deep learning methods, can overcome these limitations by leveraging sensor data to learn relationships without explicit professional knowledge. These methods have the potential to achieve better generalization performance. Specifically, statistical models estimate a tool’s RUL by fitting the observed values of the tool cutting process into random process models using probability methods. Common statistical methods include Wiener process models, random parameter models, and Markov models [6,7,8]. For instance, Sun et al. [9] utilized the Wiener process to construct a tool RUL prediction model and updated its parameters through historical and real-time data. Liu et al. [7] proposed an improved implicit semi-Markov model to describe the relationship between tool performance degradation and time to estimate the degradation state and RUL distribution of the tool. Although tool RUL prediction based on statistical methods has significant achievements, improper selection of random processes will lead to insufficient accuracy.

The machine learning-based methods and deep learning-based methods construct models by indirectly or directly learning the appropriate information in the sensor data to predict the RUL of the tool. However, the manual extraction of features from sensor data serves as a key prerequisite for constructing training samples for machine learning-based methods, which requires operators to have relevant experience in signal processing. Li et al. [2] used the support vector regression (SVR) model to construct a relationship between the sensor data and the tool wear value and employed a particle filter to update the parameters to achieve tool RUL prediction. Similarly, Gokulachandran and Mohandas [10] applied fuzzy neural regression and SVR algorithms to estimate the RUL of a tool. In fact, machine learning-based methods face challenges in extracting all the hidden information that represents the process of tool performance degradation from the sensor data. Conversely, deep learning-based methods exhibit remarkable adaptability and can automatically extract deep degradation features from sensor data. Therefore, deep learning-based methods, such as convolutional neural networks (CNNs) [11,12] deep belief networks (DBNs) [13,14], and recurrent neural networks (RNNs) [15,16], have demonstrated superior generalization performance and have been widely and successfully applied in tool RUL prediction [17,18,19]. Xu et al. [20] constructed an RUL prediction method by combining one-dimensional convolutional layers, pooling layers, residual connections, and attention mechanisms. To overcome the limited sample of fault data, Yao et al. [21] developed a transfer reinforcement learning model with long short-term memory (LSTM) and verified its higher RUL prediction accuracy through the tool dataset. In addition, to enhance the calculation speed of a time series model, Cheng et al. [22] constructed an autoencoder quasi-recurrent neural network (AEQRNN) model to learn the appropriate features from sensor data to apply in predicting tool RUL.

In the aforementioned literature, numerous scholars conducted extensive research in the field of predicting the RUL of cutting tools. However, there has been limited research focusing on simultaneously predicting both the health status of cutting tools and their RUL. When devising a maintenance plan for a tool, relying solely on its RUL is insufficient. Further prediction of the wear stages of the tool can provide a better understanding the wear trend and rate of the tool during its RUL, which contributes to developing more refined maintenance plans to enhance machining quality and reduce scrap rates. Therefore, conducting comprehensive research on the simultaneous prediction of the RUL and health status of cutting tools is significant for the predictive maintenance of cutting tools in manufacturing processes. In addition, traditional neural networks are prone to interference from noise in sensor data, resulting in lower prediction accuracy. In order to address the aforementioned issues, a multi-task joint learning model based on a transformer encoder and customized gate control (TECGC) is proposed for tool RUL prediction and wear stage prediction. The customized gate control (CGC) structure of the TECGC model can extract shared low-level features and task-specific features from tool cutting feature data, achieving complementary feature information between two tasks. This process can enable the model to more accurately capture key feature information during tool cutting, improving its generalization ability and prediction accuracy. In addition, this paper develops a multi-task dynamic adaptive loss function, which enables the model to automatically balance the RUL prediction task of the tool and the wear stage prediction task of the tool during the training process. This approach can reduce manual intervention and improve the training efficiency and stability of the model.

The remainder of this paper is organized as follows. Section 2 develops a multi-task joint learning TECGC model and a multi-task dynamic adaptive loss function. At the same time, the feature extraction and selection of the tool sensor monitoring data, as well as the hyperparameter information of the model, are presented. Section 3 introduces relevant information about the PHM2010 dataset. In Section 4, based on the PHM2010 dataset, the prediction performance of the TECGC model is discussed. Finally, the conclusions of the study are presented in Section 5.

## 2. The Proposed Method

In this section, the feature extraction and selection of sensor data during tool cutting process are introduced. The architecture and hyperparameter information of the multi-task joint learning model based on a transformer encoder and customized gate control (TECGC) model are presented. Additionally, the dynamic adaptive loss function is developed.

### 2.1. Feature Extraction and Selection

The data monitored by sensors during the tool cutting process contain a large amount of raw information, and the amount of data is relatively large. In order to reduce redundant feature data and alleviate the computational burden of the model, the statistical features of sensor monitoring data during the tool cutting process are extracted in this paper. The statistical features are presented in Table 1. Then, the Pearson correlation coefficient between features and tool wear value is calculated to remove the redundancy features. The Pearson correlation coefficient can be mathematically expressed as:(1)$$p_{f^{j},w}=\left|\frac{\sum_{i=1}^{n}\left(f_{i}^{j}-{\bar{f}}^{j})(w_{i}-\bar{w}\right)}{\sqrt{\sum_{i=1}^{n}{\left(f_{i}^{j}-{\bar{f}}^{j}\right)}^{2}\sum_{i=1}^{n}{\left(w_{i}-\bar{w}\right)}^{2}}}\right|$$
where $f_{i}^{j}$ and $w_{i}$ represent the value of the feature $f^{j}$ and the tool wear value during the *i*-th cutting operation, respectively, $p_{f^{j},w}$ represents the correlation coefficient between the feature $f^{j}$ and the tool wear value $w_{i}$, and ${\bar{f}}^{j}$ and $\bar{w}$ represent the average value of the feature $f^{j}$ and the average value of the tool wear, respectively. For $p_{f^{j},w}\in [0,\;1]$, the closer its value is to 1, the stronger the correlation between the feature and the tool wear. Conversely, as $p_{f^{j},w}$ approaches 0, the correlation becomes weaker.

In this paper, features with a correlation coefficient exceeding 0.95 are selected as sensitive features of the tool degradation process. Finally, the following features are selected: maximum value and spectral power of the cutting force in the *x*-direction, the variance, kurtosis, and spectral power of the cutting force in the *y*-direction, and the variance, maximum value, and kurtosis of the cutting force in the *z*-direction. In addition, the features are normalized by the max–min normalization method, which can be expressed as:(2)$$f_{i}^{n}=\frac{f_{i}-\min(f_{i})}{\max(f_{i})-\min(f_{i})}$$
where $f_{i}^{n}$ represents the normalized feature. The normalized features are shown in Figure 1, which demonstrates that with an increase in tool cutting cycles, the features of tool wear have a significant growth trend.

### 2.2. The Proposed TECGC Model

In order to achieve both tool RUL prediction and tool wear stage prediction simultaneously, a multi-task joint learning model based on a transformer-encoder and CGC (TECGC) is proposed in this section. As depicted in Figure 2, it is primarily composed of a transformer encoder, a CGC, a Tower-reg for tool RUL prediction, and a Tower-cla for tool wear stage prediction. Detailed information of these modules is introduced in subsequent sections. Furthermore, a multi-task dynamic adaptive loss function is developed to enhance the accuracy of the model in both the tool RUL prediction task and tool wear stage prediction task. This loss function facilitates the achievement of efficient training of the model.

#### 2.2.1. The Transformer Encoder

During the cutting process of the tool, the data monitored by the sensors constitute time series data. Effectively capturing temporal feature information in sensor data is crucial for tool RUL prediction and wear stage prediction. However, the traditional RNN models, which employ a recursive computing structure, are prone to vanishing or exploding gradients when learning from sensor data [23,24,25]. Moreover, these models tend to exhibit slower learning. In contrast, the encoder component of the transformer [26] exhibits strong parallel computing ability, enabling efficient processing of long-sequence data and avoiding the problems of vanishing and exploding gradients. It can capture long-distance dependencies and learn correlations within sensor data, thereby improving the learning speed and accuracy of the model. Therefore, the transformer encoder is employed as the backbone of the TECGC model to capture the features from sensor data to enhance its robustness and prediction performance.

As depicted in Figure 2, the transformer encoder architecture comprises several key components, as follows.


(1)Input embedding layer: This layer serves to transform the input data into a high-dimensional vector representation. This process enables the model to work in a high-dimensional representation space where relationships and patterns within the data can be more readily identified. This layer can be mathematically represented as:
(3)$$X_{output}=WX_{input}+b$$
where $X_{input}$ represents the input data, $X_{output}$ denotes the output representation of the input embedding layer, and *W* and *b* represent the weight matrix and bias vector of the input embedding layer, respectively.(2)Position encoding layer: Due to the transformer encoder lacking an inherent sequential learning mechanism, a position encoding layer is introduced to enable the model to distinguish the positional relationships among different data in the input sequence. Primarily, this layer appends a series of fixed vectors to the input data, which can be mathematically expressed as follows:
(4)$$Pe(p,2i)=\sin\left(p/{10,000}^{2i/d_{m}}\right)$$
(5)$$Pe(p,2i+1)=\cos\left(p/{10,000}^{2i/d_{m}}\right)$$
where *p* is the position of the feature in the input sequence data, *i* denotes the dimension of the feature data, and *d_m_* is the dimension of the input data.(3)Multi-head self-attention layer: This layer primarily applies multiple scaled dot-product self-attention mechanisms to capture the intricate correlations between various features across different representation spaces. This approach significantly enhances the feature representation ability of the model. Specifically, the scaled dot-product self-attention mechanism is utilized to transform input data into query matrix *Q_i_*, key matrix *K_i_*, and value matrix *V_i_.* Subsequently, it calculates the similarity between the query matrix *Q_i_* and the key matrix *K_i_* to weight the value matrix *V_i_*, thereby generating an output representation of the sensor data. The mathematical formulation of this computation can be expressed as:
(6)$$A_{i}=\mathrm{softmax}(\frac{Q_{i}{\left(K_{i}\right)}^{T}}{\sqrt{d_{a}}})V_{i}$$
(7)$$Q_{i}=XW_{i}^{q},\;K_{i}=XW_{i}^{k},\;V_{i}=XW_{i}^{v}$$
where *A_i_* represents the *i*-th output representation of the scaled dot-product self-attention mechanism, *d_a_* represents the dimension of the key matrix *K_i_*, $W_{i}^{q}$, $W_{i}^{k}$, and $W_{i}^{v}$ are the learnable parameters of query matrix *Q_i_*, key matrix *K_i_*, and value matrix *V_i_*, respectively, and *T* denotes the transpose operation.


Therefore, the output of this layer can be denoted as:(8)$$\mathrm{MHA}(X)=\mathrm{Concat}(A_{1},A_{2},\ldots ,A_{i},\ldots ,A_{h})W^{h}$$
where Concat(·) represents the concatenation operation and *W^h^* is the learnable weight parameter of linear transformation.

(4)Feed-forward neural network (FNN): The FNN is composed of two linear layers and a ReLU activation function. Its primary objective is to improve the nonlinear representation ability of the model, thereby enriching the overall feature space.(5)Residual connection and layer normalization: Residual connections play a crucial role in the transformer encoder. They can enable the model to construct a depth network structure and accelerate its training speed. Furthermore, they can effectively avoid the phenomenon of gradient vanishing by providing shortcuts for gradient flow during the backpropagation of the model. In addition, layer normalization is another crucial technique utilized in the transformer encoder. This operation of normalizing the output enhances the robustness of the model and makes it more stable to variations in input data.

#### 2.2.2. Customized Gate Control (CGC)

Multi-gate mixture-of-experts (MMOE) [27] is a common framework in multi-task joint learning tasks. The MMOE shares the underlying feature extraction module, and then each task has its own independent network to process the output of the shared feature extraction layer. However, an MMOE with only a shared feature extraction module ignores task-specific feature information, resulting in poor prediction performance of the model. To address this limitation, this paper introduces customized gate control (CGC) [28], which incorporates a shared expert system alongside two task-specific expert systems. The CGC can enable the model to extract shared features and task-specific features from tool cutting feature data, achieving complementary feature information between the two tasks and enhancing its prediction performance in predicting both tool RUL prediction and tool wear stage.

As depicted in Figure 2, the CGC structure in the TECGC model is composed of a regression expert system related to the tool RUL prediction task and the corresponding gating unit Gate-T_r_, a classification expert system related to the tool wear stage prediction task and the corresponding gating unit Gate-T_c_, and a shared expert system related to both the tool RUL prediction task and the tool wear stage prediction task. Specifically, the regression expert system, classification expert system, and shared expert system are comprised of the *m_r_*, *m_c_*, and *m_s_* expert networks, respectively. Each of these expert networks contains a linear layer and a ReLU activation function. The inputs of the regression expert system, classification expert system, and shared expert system are the outputs of the transformer encoder. Therefore, the three systems can be mathematically expressed as:(9)$$\begin{array}{c} S^{r}(x)=\mathrm{Concat}(E_{1}^{r}(x),E_{2}^{r}(x),\ldots ,E_{i}^{r}(x),\ldots ,E_{m_{r}}^{r}(x))\\ S^{c}(x)=\mathrm{Concat}(E_{1}^{c}(x),E_{2}^{c}(x),\ldots ,E_{i}^{c}(x),\ldots ,E_{m_{c}}^{c}(x))\\ S^{s}(x)=\mathrm{Concat}(E_{1}^{s}(x),E_{2}^{s}(x),\ldots ,E_{i}^{s}(x),\ldots ,E_{m_{s}}^{s}(x)) \end{array}$$
where
(10)$$\begin{array}{c} E_{i}^{r}(x)=\mathrm{Relu}(W_{i}^{r}x+b_{i}^{r})\\ E_{i}^{c}(x)=\mathrm{Relu}(W_{i}^{c}x+b_{i}^{c})\\ E_{i}^{s}(x)=\mathrm{Relu}(W_{i}^{s}x+b_{i}^{s}) \end{array}$$
where $S^{r}(\cdot )$, $S^{c}(\cdot )$ and $S^{s}(\cdot )$ represent the outputs of the regression expert system, classification expert system, and shared expert system, respectively. $E_{i}^{r}(\cdot )$, $E_{i}^{c}(\cdot )$, and $E_{i}^{s}(\cdot )$ denote the *i*-th expert network in the regression expert system, classification expert system, and shared expert system, respectively. $W_{i}^{r}$, $W_{i}^{c}$, and $W_{i}^{s}$ are the learnable weight parameters of the *i*-th expert system in the regression expert system, classification expert system, and shared expert system, respectively. $b_{i}^{r}$, $b_{i}^{c}$, and $b_{i}^{s}$ are the learnable bias parameters of the *i*-th expert system in the regression expert system, classification expert system, and shared expert system, respectively.

The Gate-T_r_ and Gate-T_c_ mechanisms are specifically designed for their respective prediction tasks. These gate mechanisms dynamically assign weights to expert systems in different task systems and expert networks in the shared expert system. This approach enables the model to learn the both specific-task feature information and the feature information shared by the two tasks. This enhances the stability of the model for both the tool RUL prediction task and the wear stage prediction task. The calculation process of the Gate-Tr can be formulated as:(11)$$G^{r}(x)=g(x)*R(x)$$
where
(12)$$g(x)=softmax(W_{g}^{r}x+b_{g}^{r})$$
(13)$$R(x)=\mathrm{Concat}([S^{r}(x),S^{s}(x)])$$
where $W_{g}^{r}$ and $b_{g}^{r}$ are the learnable weight and bias parameters of the gating unit Gate-Tr. The calculation process of the gating unit Gate-Tc is similar to Equations (11) and (12).

Finally, the tool RUL prediction and tool wear stage prediction are achieved through the Tower-reg module and Tower-cla module, respectively. Specifically, the Tower-reg module and Tower-cla module are essentially a regression layer and a classification layer, respectively. The prediction results output by the model on both tasks can be expressed as follows:(14)$${\overset{⌢}{y}}^{r}=W^{r}G^{r}(x)+b^{r}$$
(15)$${\overset{⌢}{y}}^{c}=W^{c}G^{c}(x)+b^{c}$$
where $W^{r}$ and $W^{c}$ are the trainable weight parameters of the module, respectively, and $b^{r}$ and $b^{c}$ represent the trainable bias parameters.

### 2.3. Multi-Task Dynamic Adaptive Loss Function

In the process of parameter optimization in traditional multi-task models, the trade-off parameter of each task loss function must be manually adjusted. This process is often time-consuming and inefficient, relying heavily on repeated trial and error and personal experience, thereby greatly impeding the efficiency of model parameter optimization. To overcome this limitation, a multi-task dynamic adaptive loss function is developed for parameter optimization of the TECGC model. This can enable the TECGC model to adapt to the differences between different tasks. The TECGC model focuses on two main prediction tasks: tool RUL prediction and tool wear stage prediction. Each of these tasks has its own optimization objectives.

For the tool RUL prediction task, the optimization objective of the TECGC model is to minimize the root mean square error (RMSE) loss function, which is employed to quantify the error between the predicted RUL and the actual RUL of the model. Mathematically, it can be expressed as:(16)$$L^{r}=\sqrt{\frac{1}{n}\sum_{i=1}^{n}{\left({\overset{⌢}{y}}_{i}^{r}-y_{i}^{r}\right)}^{2}}$$
where $y_{i}^{r}$ represent the actual RUL.

For the tool wear stage prediction task, the optimization objective of the TECGC model is to minimize the cross-entropy loss function, which is utilized to calculate the prediction error of the model. Mathematically, it can be expressed as:(17)$$L^{c}=-\sum_{k=1}^{n_{2}}\sum_{j=1}^{n_{1}}\sum_{i=1}^{n_{0}}y_{i,j,k}^{c}\log({\hat{y}}_{i,j,k}^{c})$$
where $y_{i,j,k}^{c}$ represents the tool wear stage label and *n*_0_, *n*_1_, and *n*_2_ represent the number of samples in different wear stages of the tool.

In this paper, in order to enable the TECGC model to achieve satisfactory prediction performance in both the tool RUL prediction task and the tool wear stage prediction task, a multi-task adaptive dynamic adjustment loss function is introduced for the optimization of model parameters. This function can be represented as:(18)$$L_{j}=\alpha_{j}L_{j}^{r}+\beta_{j}L_{j}^{c}$$
where *α_j_* and *β_j_* represent the trade-off parameters. To alleviate the burden of manually adjusting these parameters repeatedly, we introduce an adjustment weight method that dynamically determines *α_j_* and *β_j_* in Equation (18). Specifically, during the training process of model, the values of *α_j_* and *β_j_* in the current iteration are calculated based on the regression and classification loss values from the previous iteration. The calculation formula can be expressed as:(19)$$\alpha_{j}=\left\{\begin{array}{ll} 0,\; & j=1\\ \frac{L_{j-1}^{c}}{L_{j-1}^{r}+L_{j-1}^{c}},\; & j\geq 2 \end{array}\right.$$
(20)$$\beta_{j}=\left\{\begin{array}{ll} 0,\; & j=1\\ \frac{L_{j-1}^{r}}{L_{j-1}^{r}+L_{j-1}^{c}},\; & j\geq 2 \end{array}\right.$$
where *j* is the current number of iterations. Based on the multi-task dynamic adaptive loss function, the joint training of the tool RUL prediction task and the tool wear stage prediction task can improve the overall performance of the model and improve its stability and reliability. The training process of the model is shown in Algorithm 1.
**Algorithm 1** Model training process by using multi-task adaptive dynamic adjustment loss function**Preparation:**Input: Training samples $D={\left\{(x_{i},y_{i}^{r},y_{i}^{c})\right\}}_{i=1}^{n}$
Initial learning rate: *γ***Training:**     for *j* = 1, 2, …, *N* do:        if *j* = 1:        Input x into the model        Output: 
${\overset{⌢}{y}}_{i}^{r}$
$,\;{\hat{y}}_{i}^{c}$
        Calculate the loss function Equations (16) and (17)        break     else:        Input *x* into the model        Output: 
y⌢ir
, y^ic
        Calculate the *α_j_* and *β_j_* by Equations (19) and (20)        Calculate the regression and classification loss by Equations (16) and (17)        Calculate the total loss by Equation (18)        Update the model parameters *W b* by using Adam optimizer     End for**Return:**The trained model parameters: *W b*

### 2.4. The Hyperparameter Information of the Proposed TECGC Model

In this paper, based on the multi-task dynamic adaptive loss function in Equation (18), an Adam optimizer with a learning rate of 0.001 is utilized to find the optimal model parameters. In addition, in the process of model training, a batch size of 32 samples is transmitted into the model per iteration to update the parameters. The training epoch is set to 1000. The hyperparameter information of the TECGC model is summarized in Table 2.

## 3. Data Description

The PHM2010 tool wear dataset [29] contains six sub datasets, encompassing sensor data from seven channels during the working process of six different high-speed CNC milling tools. These data channels encompass cutting force signals in the *x*, *y*, and *z* directions, vibration signals in the *x*, *y*, and *z* directions, and acoustic emission signals. At the same time, the wear status of the tool is quantified through a microscope. Figure 3 illustrates the experimental platform, while Table 3 provides a detailed overview of the experimental parameters. These sub-datasets are denoted as C1, C2, C3, C4, C5, and C6. Among these, the wear of the three flutes of the tool was recorded in C1, C4, and C6. Therefore, the three datasets are selected as experimental datasets to evaluate the multi-task learning effectiveness of the TECGC model in both the tool RUL prediction task and the tool wear stage prediction task.

The wear on the three flutes of the tool is recorded and the average of these wear measurements is employed as the tool wear value. Usually, during the cutting process, the cutting tool mainly undergoes an initial wear stage, steady-state region, and accelerated wear zone. According to [30], when $VB^{\prime \prime }<0$, the tool is in the initial wear stage. Otherwise, it is in the significant wear stage. The significant wear stage can be further divided into the steady-state region and accelerated wear zone. In order to better distinguish the last two stages of the tool cutting process, a tool wear increment of 0.003 is set as a threshold. The steady-state region of the tool is when $VB^{\prime \prime }>0$ and the wear increment is less than or equal to 0.003. Then, the subsequent stage is the accelerated wear zone. It is worth noting that before calculating the wear increment of the tool, it is necessary to normalize its wear value to [0, 1]. Finally, the detailed division of the wear stages of the three tools is shown in Table 4, where the values represent the cutting cycles.

Figure 4 presents the average wear value of tool C4 and its three wear stages. The horizontal axis denotes the cutting cycles of the tool, while the vertical axis represents the normalized average wear. Following the 37th cutting cycle, the $VB^{\prime \prime }$ of the tool C4 shifts from negative to positive. Therefore, during the initial 37 cutting cycles, tool C4 is in the initial wear stage. During this stage, the surface roughness of the tool is high, the contact area with the workpiece is small, and the cutting force is relatively low, but with the development of tool side wear, the cutting force continues to increase. In addition, the wear speed of the tool is faster during this stage, but the wear value is relatively small. Subsequently, when the 219 cutting cycles are completed, the average wear increment of the tool reaches 0.0031, so tool C4 is in the steady-state region during the 38th to 218th cutting cycles. From the depiction in Figure 4, it is evident that during the accelerated wear zone of the tool, both its wear speed and wear value sharply increase, which significantly impacts the surface quality of the processed workpiece and potentially leads to tool breakage. Therefore, accurate prediction of the tool wear stage is crucial for industrial production. In addition, this article assigns the values 0, 1, and 2 as the corresponding label values for the initial wear stage, steady-state region, and accelerated wear zone of the tool. At the same time, a sliding time window with a length of 10 and a sliding step of 1 is employed to extract samples from feature data. Ultimately, the dimension of the model sample is 305 × 10 × 8. In this paper, two datasets were selected from C1, C4, and C6 as the training dataset and the other as the testing dataset to analyze the predictive performance of the TECGC model. The specific division of training and testing sets is shown in Table 5.

## 4. Experimental Study and Discussion

In this section, the effectiveness of the TECGC model in both the tool RUL prediction task and tool wear stage prediction task is evaluated on the PHM2010 dataset. The experiments were performed using an Nvidia GeForce RTX 3060s with Pytorch 1.10.0.

### 4.1. Evaluation Index

In this paper, the accuracy of the TECGC model in the tool RUL prediction task is evaluated by the RMSE and score function, which serves as a benchmark for comparing the effectiveness of the TECGC model with existing models. RMSE is used to quantify the prediction error in an RUL prediction task. Notably, in the RUL prediction task, if the predicted RUL of the model exceeds the actual RUL, it results in a delayed prediction of RUL. In the practical cutting process, delayed prediction of tool RUL may lead to the use of damaged tools for manufacturing, subsequently impacting the surface quality of the manufactured product. In contrast to the early prediction of RUL, the score function imposes a greater penalty on the delayed prediction of RUL. Their calculation formulae can be denoted as:(21)$$\mathrm{RMSE}=\sqrt{\frac{1}{N}\overset{N}{\underset{i=1}{\sum }}{\left({\overset{⌢}{y}}_{i}^{r}-y_{i}^{r}\right)}^{2}}$$
(22)$$\mathrm{Score}=\overset{N}{\underset{i=1}{\sum }}s_{i},s_{i}=\left\{\begin{array}{c} e^{-\frac{{\overset{⌢}{y}}_{i}^{r}-y_{i}^{r}}{13}}-1,\;{\overset{⌢}{y}}_{i}^{r}-y_{i}^{r}\leq 0\\ e^{\frac{{\overset{⌢}{y}}_{i}^{r}-y_{i}^{r}}{10}}-1,\;{\overset{⌢}{y}}_{i}^{r}-y_{i}^{r}>0 \end{array}\right.$$
where *N* is the number of samples.

In addition, the accuracy index is utilized in this paper to evaluate the precision of the model in the tool wear stage prediction task. The calculation formula is as follows:(23)$$A\mathrm{ccuracy}=\frac{n_{p}}{N}$$
where *n_p_* represents the number of tool wear stages results consistent with the actual results.

### 4.2. Prediction Performance Analysis

In this paper, a TECGC model is proposed for both the tool RUL prediction task and the tool wear stage prediction task. The model is trained jointly on these two tasks using a developed multi-task adaptive dynamic adjustment loss function. Additionally, based on RMSE, score, and accuracy, the prediction performance of the TECGC model is evaluated on both the tool RUL prediction task and the tool wear stage prediction task.

#### 4.2.1. Prediction Performance Evaluation of Multi-Task Learning Model

Table 6 presents the RMSE and score of the TECGC model in the tool RUL prediction task. The results demonstrate that the TECGC model achieves superior RUL prediction accuracy compared to other models, including traditional machine learning models, such as the Paris model [31] and original exponential model [31], and deep learning models, such as the HAGCN model [32], the CNN-LSTM model [33], and the GAM-CapsNet model [34]. Additionally, the prediction performance of the TECGC model in the tool wear stage prediction task is analyzed using an accuracy index. The TECGC model achieves a prediction accuracy of 0.953%, 0.967%, and 0.979% for the C1, C4, and C6 wear stage prediction tasks, respectively. Overall, the TECGC model exhibits both high-precision tool RUL prediction and high-precision tool wear stage prediction.

Figure 5 illustrates the comparison between the predicted RUL and the actual RUL of TECGC in the tool RUL prediction task alongside the confusion matrix diagram in the tool wear stage prediction task. The comparison reveals that the predicted RUL of the TECGC model is close to the actual RUL for tools C1, C4, and C6. Moreover, the model exhibits a remarkable capacity to accurately predict the trend of tool performance degradation. The horizontal axis of the confusion matrix diagram of the prediction results of TECGC in the tool wear stage prediction task represents the tool wear stage category identified by the TECGC model, while the vertical axis represents its actual category. Figure 5 shows that the TECGC model can excellently identify the wear stage of the tool on C1, C4, and C6 tools. In summary, Figure 5 demonstrates that the TECGC has robust performance in tool health prognosis, exhibiting both precise RUL prediction and accurate wear stage identification.

Figure 6 presents a comparison of the prediction accuracy between the TECGC model and other models, such as CNN-CNN, LSTM-LSTM, DNN-DNN, and CNN-LSTM models [32], in the tool wear stage prediction tasks. The results clearly indicate that the TECGC model exhibits significantly superior prediction accuracy compared to the other models. This demonstrates that the TECGC model not only achieves satisfactory prediction results in the tool RUL prediction task but also has high-precision prediction ability in the tool wear stage prediction task. Therefore, the proposed TECGC model has the potential to help operators schedule more detailed maintenance plans for different wear stages of tools, thereby improving production efficiency and reducing maintenance costs. For example, during the initial wear stage of the tool, the lifespan of the tool can be prolonged by adjusting production speed. In the normal wear stage, production efficiency can be enhanced by increasing production speed. Moreover, in the rapid wear stage, the decision to replace the tool can be informed based on the predicted tool RUL value to ensure product quality. The high-precision prediction of tool wear stages can make precise production control, leading to optimized production processes and resource utilization.

#### 4.2.2. Effectiveness Analysis of Model Structure

To validate the efficiency of the TECGC model structure, a comparative analysis of its prediction performance is conducted with several variant models, as follows.

LSTMCGC: This variant replaces the transformer encoder module of the TECGC model with the two LSTM layers, each comprising 64 neurons.

GRUCGC: In this variant, the transformer encoder module of the TECGC model is substituted with the two GRU layers, each comprising 64 neurons.

CNNCGC: This variant replaces the transformer encoder module of the TECGC model with two one-dimensional CNN layers, each equipped with 64 filters of size 3.

In order to ensure fairness in the comparison of model prediction performance, joint training on the tool RUL prediction task and tool wear stage prediction task of the above models is conducted based on the same hyperparameters and data samples. The predictive performance evaluation index values of these models are presented in Table 7. The result demonstrates that the TECGC model exhibits competitive prediction accuracy in both tool RUL prediction tasks and tool wear stage recognition tasks, significantly outperforming other variant models. This comparison highlights the effectiveness of the TECGC model structure and exhibits its potential as a powerful tool for tool condition monitoring and maintenance planning.

#### 4.2.3. Effectiveness Analysis of Dynamic Adaptive Loss Function

In order to mitigate the reliance on manual tuning of the trade-off parameters in the loss function, a multi-task dynamic adaptive loss function is developed for training the TECGC model. This adaptive loss function enables the training process to dynamically adjust to balance the relationship between the tool RUL prediction task and the tool wear stage prediction task. This dynamic adjustment is achieved by calculating the error value of the previous iteration’s RUL prediction and the classification error of the tool wear stage. To validate the effectiveness of this loss function, the prediction accuracy of the TECGC model trained with the proposed function is compared with that of the TECGC model trained using various manually tuned parameter combinations (*α_j_*, *β_j_*) for both the RUL prediction task and the tool wear stage prediction task of tool C6.

The results presented in Figure 7 clearly demonstrate the effectiveness of the multi-task dynamic adaptive loss function in enhancing the prediction performance of the TECGC model for both the RUL prediction task and the tool wear stage prediction task for tool C6. Furthermore, the model also exhibited the highest prediction accuracy in the wear stage prediction task for tool C6, highlighting its ability to achieve precise tool wear prediction. When *α_j_* = 0.7 and *β_j_* = 0.7, the prediction performance of the TECGC model remains acceptable, but the values of these two parameters are required to be determined through multiple training based on different parameter combinations. This would prolong the model training. In contrast, the dynamic adaptive loss function can achieve the optimal predictive performance of the model after training it once. Therefore, its computational efficiency is higher. In summary, the dynamic adaptive loss function is suitable for the goal of multi-task learning, as it not only improves prediction performance in both the RUL prediction task and the tool wear stage prediction task of the model but also effectively mitigates the reliance on manual parameter tuning, ultimately enhancing the efficiency of the model training process.

## 5. Conclusions

The RUL of a tool and its wear status are crucial for the intelligent maintenance of the tool. Accurately predicting tool RUL and identifying its wear stage contribute to helping operators develop detailed maintenance strategies, make more detailed production plans for CNC machines, and thus improve production quality and efficiency. In this paper, based on a transformer encoder and CGC structure, a multi-task joint learning TECGC model is proposed for tool RUL prediction and wear stage prediction. In addition, a dynamic adaptive loss function is developed to implement the training process of the TECGC model to reduce manual intervention to the trade-off parameters.

Based on the PHM2010 dataset, the effectiveness of the TECGC model in tool RUL prediction tasks and tool wear stage recognition tasks was conducted. The experimental results show that the TECGC model achieves both high-precision tool RUL prediction and high-precision tool wear stage prediction. In the task of tool RUL prediction, the predictive performance of the TECGC model was compared with the state-of-the-art models. The results demonstrate that the TECGC model has strong competitiveness. In addition, compared with other loss parameter combinations, the dynamic adaptive loss function can avoid the process of manual trial and error in selecting parameters, effectively improving the training efficiency.

Overall, the TECGC model has shown great potential for application in the field of intelligent tool maintenance. However, the model suffers from insufficient interpretability. In the future, physical information during tool cutting processes should be introduced to enhance the interpretability of neural networks.

## Figures and Tables

**Figure 1 sensors-24-04117-f001:**
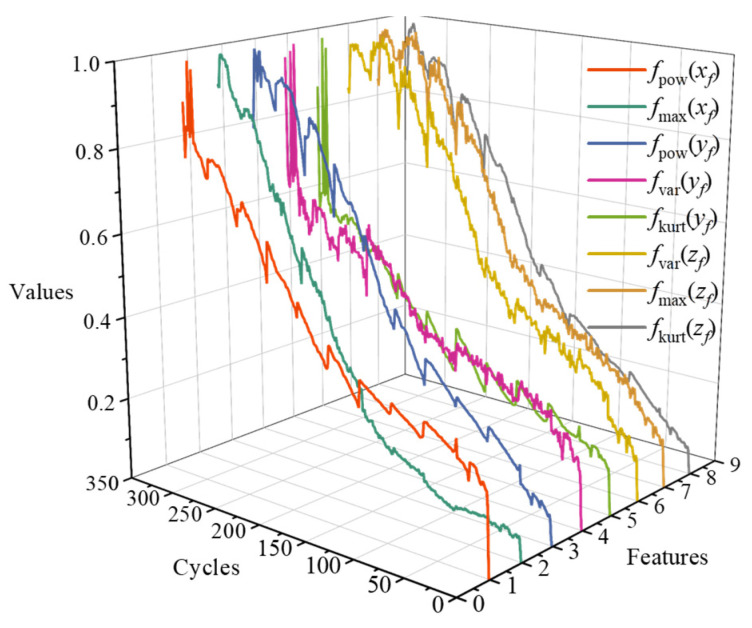
The normalized features.

**Figure 2 sensors-24-04117-f002:**
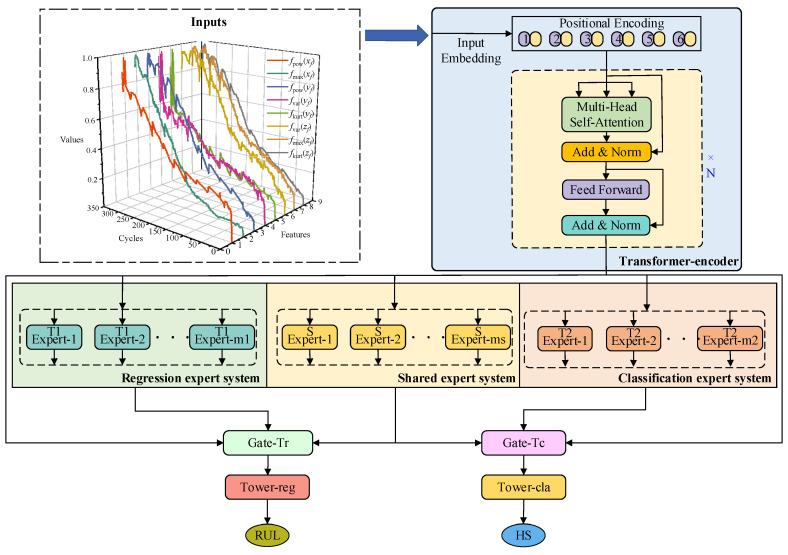
The architecture of the TECGC model.

**Figure 3 sensors-24-04117-f003:**
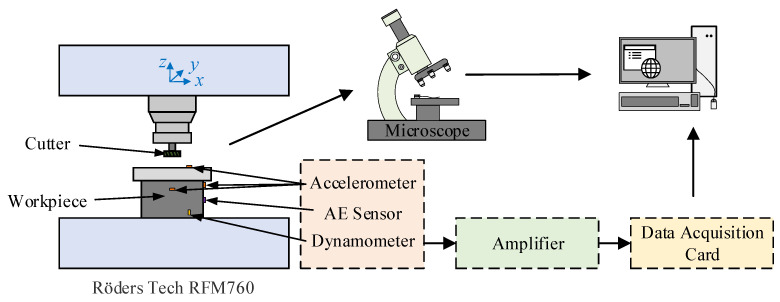
The experimental platform of high-speed CNC.

**Figure 4 sensors-24-04117-f004:**
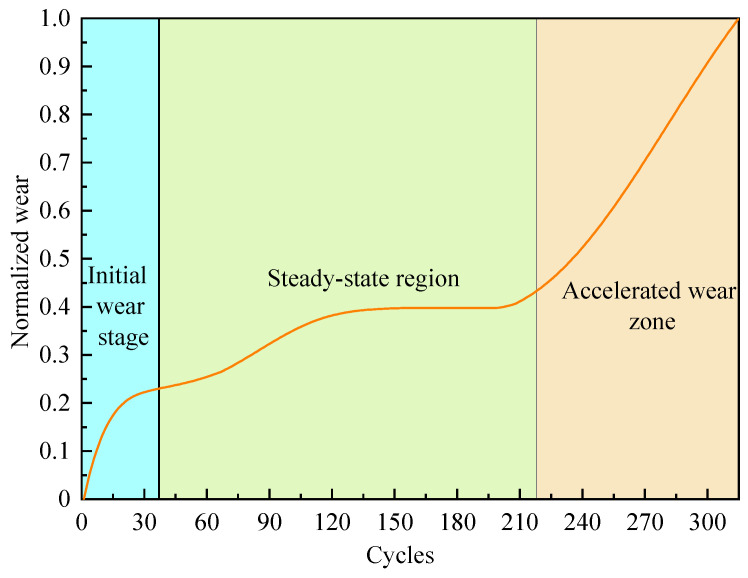
The wear stage of tool C4.

**Figure 5 sensors-24-04117-f005:**
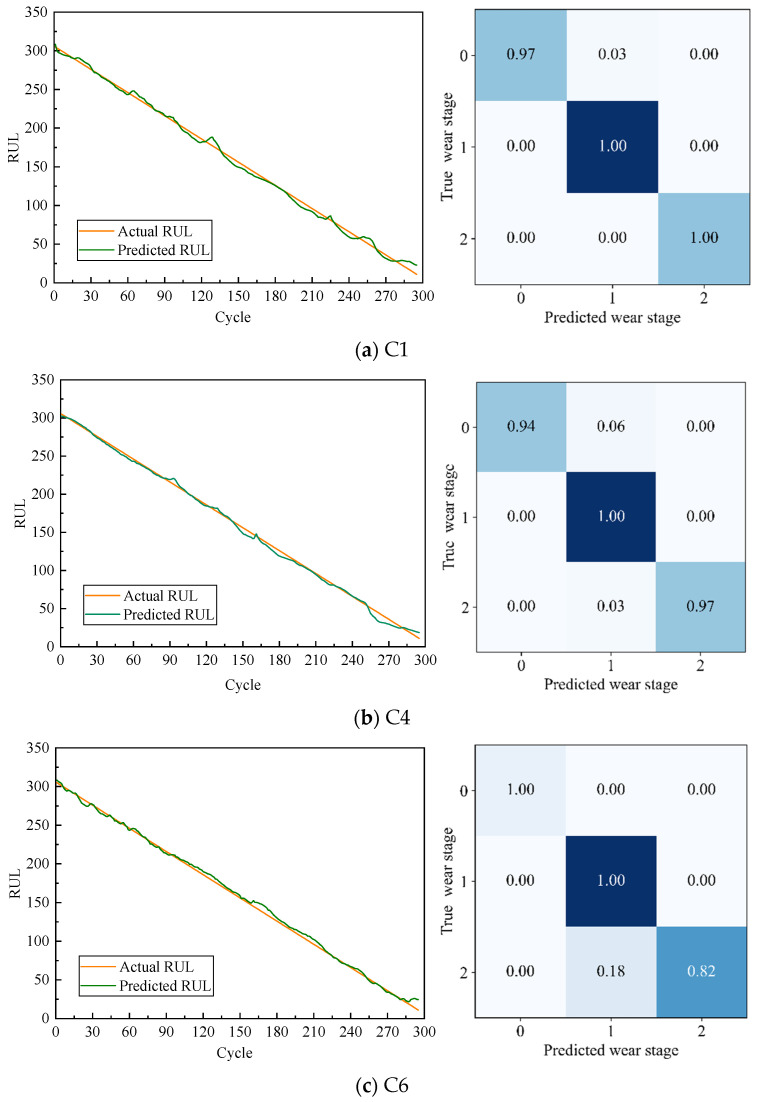
Tool RUL prediction results and wear stage prediction results.

**Figure 6 sensors-24-04117-f006:**
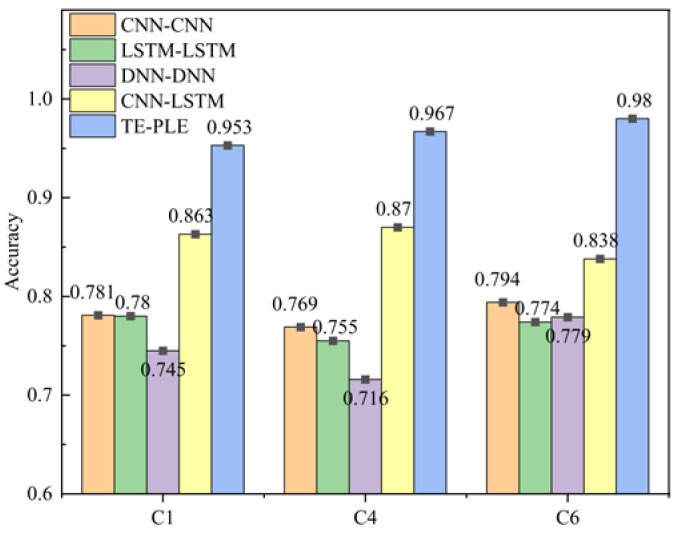
Prediction precision among the CNN-CNN, LSTM-LSTM, DNN-DNN and PE-CGC models.

**Figure 7 sensors-24-04117-f007:**
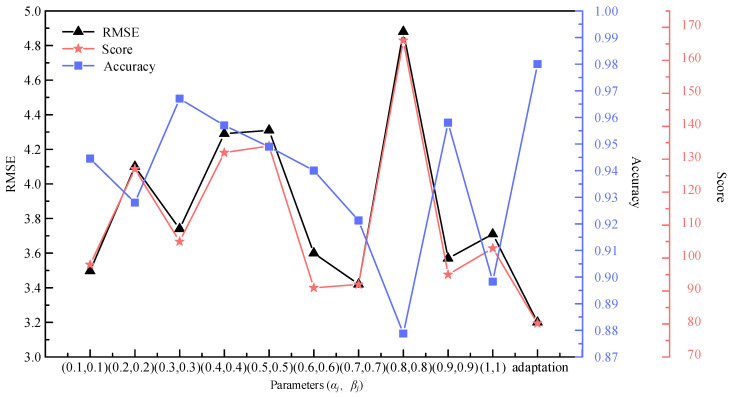
Comparison of prediction performance of models under different combinations of *α_j_* and *β_j_*.

**Table 1 sensors-24-04117-t001:** Calculation formulae of features.

Features		Equation	Features		Equation
Time Domain	Root mean square	$f_{rms}=\sqrt{\frac{1}{n}\sum_{i=1}^{n}x_{i}^{2}}$	Frequency Domain	Spectral skewness	$f_{speskew}=\sum_{i=1}^{n}{\left(\frac{F_{i}-\bar{F}}{\sigma }\right)}^{3}S(F_{i})$
	Variance	$f_{\mathrm{var}}=\frac{1}{n}\sum_{i=1}^{n}{\left(x_{i}-\bar{x}\right)}^{2}$		Spectral kurtosis	$f_{kurtt}=\sum_{i=1}^{n}{\left(\frac{F_{i}-\bar{F}}{\sigma }\right)}^{4}S(F_{i})$
	Maximum	$f_{\max}=\max(x)$		Spectral power	$f_{\mathrm{pow}}=\sum_{i=1}^{n}F_{i}^{3}S(F_{i})$
	Peak-to-peak	$f_{pp}=\max(x)-\min(x)$	Time–frequency domain	Wavelet energy	$f_{wave}=\sum_{i=1}^{n}\frac{wt_{\varphi }^{2}(i)}{n}$
	Skewness	$f_{skew}=E[{\left(\frac{x-\mu }{\sigma }\right)}^{3}]$			
	Kurtosis	$f_{kurt}=E[{\left(\frac{x-\mu }{\sigma }\right)}^{4}]$			

**Table 2 sensors-24-04117-t002:** Hyperparameters of the TECGC model.

Module	Parameter	Value
Transformer encoder	The number of encoders	1
	The number of heads	4
	*d_a_*	32
CGC	Units	64
	The number of experts in the expert-task 1	4
	The number of experts in the expert-task 2	4
	The number of experts in the expert-shared	4
Tower-reg	The number of neurons	1
Tower-cla	The number of neurons	3

**Table 3 sensors-24-04117-t003:** Parameters of tool wear experiment.

Parameter	Value
Spindle speed	10,400 rpm
Tool feed rate	1555 mm/min
Workpiece material	HRC52 stainless steel
Tool material	high-speed steel
Sampling frequency	50 k HZ
The cutting depth of the tool in the y-direction	0.125 mm
The cutting depth of the tool in the z-direction	0.2 mm

**Table 4 sensors-24-04117-t004:** The wear stage of the three tools.

Tool	Initial Wear Stage	Steady-State Region	Accelerated Wear Zone
C1	[1, 79]	[80, 247]	[248, 315]
C4	[1, 37]	[38, 218]	[219, 315]
C6	[1, 23]	[24, 192]	[193, 315]
Label	0	1	2

**Table 5 sensors-24-04117-t005:** Training and testing datasets of the model.

Task	Training Dataset	Test Dataset
Task 1	C1 + C4	C6
Task 2	C1 + C6	C4
Task 3	C4 + C6	C1

**Table 6 sensors-24-04117-t006:** The evaluation indexes of the TECGC model in the tool RUL prediction task.

Model	C1		C4		C6	
	RMSE	Score	RMSE	Score	RMSE	Score
Paris model [31]	16.37	-	27.84	-	25.02	-
Original exponential model [31]	14.46	-	29.73	-	24.1	-
HAGCN [32]	23.2	2278.3	11.2	589.6	15.6	1162.8
CNN-LSTM [33]	44.7	46,929	15.1	829	31.7	19,966
GAM-CapsNet [34]	4.59	148	7.44	221	5.99	217
TECGC	4.09	126	4.31	113	3.20	80

**Table 7 sensors-24-04117-t007:** The prediction performance of the TECGC model with other variant models.

Model	C1			C4			C6		
	RMSE	Score	Accuracy	RMSE	Score	Accuracy	RMSE	Score	Accuracy
LSTMCGC	6.84	225	0.872	4.70	154	0.950	4.66	105.10	0.908
GRUCGC	7.897	258.5	0.908	4.49	136	0.964	4.318	102.23	0.913
CNNCGC	8.25	303	0.890	4.42	123	0.944	4.10	101.56	0.890
TECGC	4.09	126	0.953	4.31	113	0.967	3.20	80	0.979

## Data Availability

Data will be made available on reasonable request.

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
