# Peer review of "A Multi-Task Joint Learning Model Based on Transformer and Customized Gate Control for Predicting Remaining Useful Life and Health Status of Tools†"

_sensors, 2024, doi:10.3390/s24134117_

Round 1

Reviewer 1 Report

Comments and Suggestions for Authors

In this paper, a multi-task joint learning TECGC model is proposed for tool RUL prediction and wear stage prediction. A dynamic adaptive loss function is developed to implement the training process of the TECGC model. The topic of this paper is interesting, and it is well-organized. Specific comments are as follows:

1. This study focused on predicting the remaining useful life (RUL) of tools as dependent processes. Why is there a correlation between them?

2. A multi-task joint learning model based on a Transformer-encoder and customized gate control (TECGC) is proposed for simultaneous prediction of tool RUL and tool wear stages. What are the advantages of the proposed model?

3. A dynamic adaptive multi-task learning loss function is proposed for the model training to enhance its calculated efficiency. What indicators quantitatively describe the analysis of computational efficiency?

4. In this study, how to choose the tradeoff parameters of the loss function reasonably? The authors should provide a detailed explanation.

5. More recent works about remaining useful life prediction should be included in this paper, like “A novel machine learning method for multiaxial fatigue life prediction: Improved adaptive neuro-fuzzy inference system. International Journal of Fatigue, 2024, 178: 108007.”

6. The effectiveness of the TECGC model is evaluated using the PHM2010 dataset. Suggest the authors to compare the prediction results of this method with other machine learning methods.

Reviewer 2 Report

Comments and Suggestions for Authors

This paper presents a combined solution for cutting tool RUL prediction and classification of the state of tool wear using a multi-task joint model based on transformer architecture and customized gate control. The idea is quite creative, and the paper is a good read. However, there are  some issues that need to be addressed before this manuscript is ready to be published:

1.      The authors unite the RUL prediction and wear level classification problem in one big network. Would it be more computationally economic to divide these problems into two separate networks? Is it fair to say that the much simpler classification problem requires a complex MMOE solution and causes the regression expert system to look for tradeoffs, which can potentially degrade its performance? 

2.      The input embedding step requires more details. Additionally, more discussion on providing the sequence information of the data pieces to the transformer encoder is needed, as it is probably the weakest point of the paper. Recently, multiple papers discussing positional encoding have credited it with providing great performance capabilities when dealing with natural language data. However, for time-series data, where complex non-linear patterns and dependencies are present, simple positional encoding might be insufficient, and more sophisticated encoding methods would sometimes be preferable.

3.      A very serious question is the rationale for manual feature extraction and selection. The authors do not provide any explanation except for “Due to the large amount of data collected from each sampling point…”. This step requires thorough discussion.

4.      Some sections need rearrangement. For example, Section 3 “The hyperparameter information of the proposed TECGC model” can be just subsection 2.4. Section 4 is too overloaded with information and needs to be split into several sections: it contains the description of the dataset, a description of the feature extraction and selection process, performance evaluation metrics, performance evaluation for classification, performance evaluation for RUL prediction, comparison with the reference methods and demonstrations of the analysis of the adaptive loss function. I would recommend placing the dataset description and assignment of the wear stages in a separate section. Feature extraction and selection in a separate section and possibly before describing the transformer-encoder in subsection in 2.1. It will help to present some step-by-step logical workflow.  Finally, all performance evaluation-related subsections can be placed in one more separate section right before the conclusion.

5. The text in Figure 1 is very small or even hidden behind the images.

6.   For the wear level classification problem, in Section 4.2, the authors assign three categories of wear according to the methodology in reference [30]. This neat and logical method designates three categories of wear development: the initial wear stage, the steady state region, and the accelerated wear zone. However, in the text, the authors use the term “Normal wear stage” for the “Steady-state region.” The "Normal wear stage" is usually determined by the absolute value of the flank wear according to ISO standards, and the wear designation method chosen by the authors is related to the character of wear development. Assignment of the wear stages using these methods does not necessarily yield the same divisions. For this reason, I advise the authors to use the same terminology as in reference [30].

7.      In Section 4.2. there is the following statement: “…tool C4 is in the initial wear stage. During this stage, due to its high surface roughness and small contact area with the workpiece, the cutting force is relatively high.” In reality, at this stage, the cutting forces are relatively low and only grow along with the development of flank wear, which can be also seen from the experimental data used in this work. Even in Figure 3 of this manuscript some of the features selected as the most wear-sensitive (maximum value and spectral power of the cutting force in the x-direction, spectral power of the cutting force in the y-direction, maximum cutting force in the y-direction) show almost linear increase as the number of cutting cycles rises. Similarly, the statement “In this stage, as the number of cutting cycles increases, the contact area between the tool and the workpiece gradually expands, and the cutting force gradually decreases, resulting in a relatively slow wear rate of the tool.” needs to be revised as well. In the initial stage of wear, it is not the cutting force, but the stress resulted on the workpiece is high due to the sharp edge of the tool and small contact area.

8.      Section 4.4.1. says: “In this paper, the training set consists of data from two tools in C1, C4, and C6, while the remaining data is used as the test set.” Leaving the reader to presume that the testing is done on C2, C3, and C5. However, later in that section: “The calculation shows that the model achieves a prediction accuracy of 0.953%, 0.967%, and 0.979% for the C1, C4, and C6 wear stage prediction tasks, respectively.” “The comparison reveals that the predicted RUL of the TECGC model is close to the actual RUL for tools C1, C4, and C6.” These statements create confusion on which data was used and for what. Also, no information is provided on how the data is split for training/testing/validation.

9.      In Section 4.4.2.  for the performance comparison, the authors replace the TECGC model structure with the LSTM layers, GRU layers, and CNN layers. The question is:  What is the input data in this case and what are the sizes of input layers? Are manual feature extraction and selection involved as in the proposed method?  If it is, would it be possible to provide the reference methods’ results on raw data? Please, clarify this. Because 14.2% accuracy difference with the closest performing reference method on the C6 case looks suspicious.

10.  Section 4.4.3. needs more details and axis titles for Figure 7 should be added.

Comments on the Quality of English Language

The English language is fine.

Reviewer 3 Report

Comments and Suggestions for Authors

please see attachment.

Round 2

Reviewer 2 Report

Comments and Suggestions for Authors

The authors have satisfactorily addressed all the major concerns.

Reviewer 3 Report

Comments and Suggestions for Authors

Thank you, all the questions and suggestions are well addressed. Best of luck from my side.